# The Effects of Aging on Microstructures and Rheological Properties of Modified Asphalt with GO/SBS Composite

**DOI:** 10.3390/polym16111504

**Published:** 2024-05-25

**Authors:** Haiwei Xie, Yixuan Jia, Weidong Liu, Zhipeng Huang, Hanyu Wang, Zuzhong Li, Chunsheng Zhu

**Affiliations:** 1School of Traffic & Transportation Engineering, Changsha University of Science & Technology, Changsha 410114, China; xiehaiwei@xjau.edu.cn; 2School of Traffic & Logistics Engineering, Xinjiang Agricultural University, Urumqi 830052, China; 320212702@xjau.edu.cn; 3Key Laboratory of Highway Engineering Technology and Transportation Industry in Arid Desert Regions, Urumqi 830000, China; 4Guangxi Key Laboratory of Road Structure and Materials, Guangxi Transportation Science and Technology Co., Ltd., Nanning 530007, China; 5School of Materials Science and Engineering, Chang’an University, Xi’an 710064, China; 2019902853@chd.edu.cn (Z.H.); 2023131075@chd.edu.cn (H.W.); zuzhongli@chd.edu.cn (Z.L.); 6Xinjiang Transportation Investment (Group) Co., Ltd., Urumqi 830000, China; zcsbest@163.com

**Keywords:** graphene oxide, styrene–butadiene–styrene-modified asphalt (SBS-MA), aging, rheological properties, FTIR (Fourier-transform infrared spectroscopy), AFM (atomic force microscopy)

## Abstract

This work aimed to investigate the effects of aging on the microstructures and rheological properties of modified asphalt with a GO/SBS composite, since the styrene–butadiene–styrene block copolymer is potentially compatible with graphene oxide (GO). The GO/SBS composites, which were used as a kind of modifier, were prepared via the solution-blending method. GO/SBS composites with varying GO contents were employed to prepare the GO/SBS-compound-modified asphalt (GO/SBS-MA). Then, the GO/SBS-MA underwent PAV (pressure aging vessel) or UV (ultraviolet) aging tests to simulate different aging circumstances. The microstructures of the asphalt binders were studied using FTIR (Fourier-transform infrared spectroscopy) and AFM (atomic force microscope) tests. Moreover, DSR (dynamic shear rheometer) and BBR (bending beam rheometer) experiments were carried out to investigate the rheological properties of the GO/SBS-MA. The results showed that the addition of GO improved the high-temperature stability of the asphalt binder while slightly impairing its performance at low temperatures. GO restrained the formation of carbonyl and sulfoxide groups as well as the breakdown of C=C bonds in the polybutadiene (PB) segment, promoting the anti-aging performance of GO/SBS-MA. Furthermore, the interactions between the GO/SBS and the asphalt binder resulted in the formation of needle-like aggregates, enhancing the stability of the asphalt binder. The asphalt binders with a higher content of graphene oxide (GO) exhibited not only a better high-temperature performance, but also a better aging resistance. It was concluded that the macroscopic properties and microstructures were significantly affected by GO, and a moderate increase in the amount of GO could contribute to a better aging resistance for GO/SBS-MA.

## 1. Introduction

Due to its superior high- and low-temperature performance, styrene–butadiene–styrene-modified asphalt (SBS-MA) has occupied more than 80% of the market shares of road petroleum asphalt in recent years [1,2]. However, the service performance of SBS-MA in pavement is strongly affected by heat, light, oxygen, axle loads, and other factors, resulting in the degradation of polymers, damage to the polymer phase, and the oxidation of the base asphalt [3,4,5]. With the prolonging of service time, the performance of asphalt binders gradually deteriorates owing to aging [4,6]. Therefore, studies on effective ways to slow down the aging process of SBS-MA have attracted the attention of many researchers.

Recently, it was reported that nanomaterials have a significant effect on preventing asphalt aging [7,8]. For instance, isophorone diisocyanate-layered double hydroxide can generate an appropriate network structure in SBS-MA, promoting the thermal oxidative and photo-oxidative aging resistance of SBS-MA [9]. Nano Al_2_O_3_ increases the number of interactions between the asphalt binder and the polymer, and improves the aging resistance of polymer-modified asphalt (PMB) [10]. Carbon nanotubes (CNTs) can reduce the occurrence of thermal cracking as well as the oxidative aging susceptibility, resulting in attainable benefits in terms of the long-term performance of the asphalt binder [11]. In addition, the graphene sheets studied by Li et al. [12] prevented oxygen from diffusing into the asphalt, thus enhancing the anti-aging properties of the asphalt binder. Graphene oxide (GO), as a carbon-based two-dimensional nanomaterial, is known to have superior advantages for improving the mechanical properties and thermal stability of polymers [13,14]. It can be utilized as a reinforcing agent in a variety of polymers to allow for abundant surface-oxygen-containing functional groups, such as hydroxyl, carboxyl, and epoxy. In addition, the chemical structure of GO is similar to that of partial bitumen components, resulting in a better compatibility between GO and asphalt compared to other additives [15,16]. Meanwhile, GO acts as an excellent gas barrier, effectively inhibiting the entry of oxygen and the release of volatile organic compounds, and thereby delaying the aging of asphalt [17]. Therefore, GO is a highly promising additive for modifying asphalt.

Considering the advantages noted above, the research on GO-modified asphalt has become a hot issue [18]. It has been observed that the effect of GO on asphalt can be mainly attributed to physical modifications [19,20]. Li et al. [21] demonstrated by tests that the lamellar structure of GO is fully stripped and then GO is evenly scattered in asphalt, promoting the performance of the asphalt. Zeng et al. [22] found that GO, as a stand-alone modifier, improved the high-temperature stability of asphalt by forming an intercalation structure. Liu et al. [20] discovered that excellent high-temperature properties of asphalt can be achieved by incorporating GO, but the low-temperature performance of the asphalt was compromised. So, a single modification by GO cannot balance the high- and low-temperature performance of an asphalt binder well.

Additionally, a few studies have reported on the enhancement effect of GO on the performance of SBS-MA. As a reinforcing agent, GO can remarkably improve the aging resistance of asphalt mixtures, owing to its ability to reduce the degradation of SBS modifiers and change the crosslinked composite structure of asphalt [23,24]. Favourable high-temperature rheological properties were also obtained for the asphalt binder by doping SBS with GO to form a composite modifier [25]. However, GO is prone to agglomeration due to its stacking effect, impairing its dispersion in the bitumen base [26]. Also, the improvement in the performance of SBS-MA mainly relies on the stability and effectiveness of the SBS structure. To achieve a better dispersion of GO in asphalt and maximize the effect of GO on enhancing the anti-aging properties of asphalt, the combination of the SBS polymer and GO should be preferentially considered. Interestingly, according to the research conducted by Yang et al. [27], PS/GO composites have been successfully prepared by the solution-blending method, and most of the GO sheets can be dispersed in the PS matrix as tactoids with several layers. Therefore, it is meaningful to adopt the solution-blending method to prepare better GO/SBS composites for asphalt modification.

The objective of this work was to investigate the effect of GO/SBS composites on the aging resistance of asphalt binders. Firstly, GO/SBS composites were prepared by means of a solution-blending procedure to achieve a homogeneous dispersion of GO in the SBS. Secondly, GO/SBS composites with a varying GO content were employed to prepare the GO/SBS-MA. PAV (pressure aging vessel), and UV (ultraviolet) aging tests were performed to simulate different aging circumstances. FTIR and AFM tests were adopted to examine the microstructural changes in the GO/SBS-MA under various aging circumstances. Finally, DSR and BBR tests were used to evaluate the rheological properties of the GO/SBS-MA before and after aging, and the anti-aging mechanism of GO/SBS-MA is also discussed.

## 2. Materials and Methods

### 2.1. Materials

The base asphalt (A-70) for the GO/SBS-MA was produced by SK Corp. (Seoul, Republic of Korea), and its physical properties are shown in Table 1. The GO powder as a raw material in this work was obtained from Guopeng Technology Company Co., Ltd. (Huizhou, China), and its basic parameters are described in Table 2. SBS T6302 (manufactured by Dushanzi Petrochemical Co., Ltd., (Dushanzi, China) was selected as the polymer matrix for the preparation of the composite materials. The technical properties of SBS T6302 are listed in Table 3.

### 2.2. Preparation of GO/SBS Composites and GO/SBS-MA

In this research, the preparation procedure of the GO/SBS composite modifiers was divided into three steps. Firstly, GO powder and SBS were blended into a cyclohexane solution using a mixer, with a weight ratio between the GO/SBS composite and cyclohexane of 1:5. Subsequently, the mixed solution was evenly subjected to an ultrasonic disperser to achieve a good dispersion of GO in the SBS. Secondly, the mixed solution was poured into a flask and stirred at 50 °C until it formed a sort of viscous liquid. After the cyclohexane solution evaporated partially, the remaining black viscous liquid was poured into a flat mould and then placed in a drying oven at 50 °C for 24 h. Finally, the GO/SBS composites were taken out of the flat mould. Following this procedure, three kinds of GO/SBS composites containing different GO contents, accounting for 0.5, 1.0, and 1.5% of the total mass of the composite modifiers, were prepared successfully [29].

The prepared GO/SBS composites were used to prepare the GO/SBS-MA. Figure 1 depicts the preparation procedure of the modified asphalt; the raw materials for the modified asphalt are listed in Table 4. According to the varying contents of GO, at 0.5, 1.0, or 1.5%, the asphalt samples were named MA-2, MA-3, and MA-4, respectively. SBS-MA without GO was used as a control group and was named MA-1. Additionally, compatibilizers primarily containing saturated and aromatic fractions were added with the modifier, contributing to the swelling of the modifier and the formation of a stronger network structure through the self-crosslinking of modifier chains. In addition, stabilizers mainly composed of sulphur were added before the end of shearing for 15 min. The stabilizers improved the resistance to segregation between the modifier and the asphalt, and enhanced the thermal stability of the modified asphalt.

### 2.3. Aging Tests

It is well known that thermal oxidation and photo-oxidation are the principal causes of asphalt aging. A pressure aging vessel (PAV) test and an ultraviolet (UV) aging test were designed, respectively.

#### 2.3.1. PAV Test

In the PAV test, asphalt samples were aged in a vessel to simulate the long-term thermal oxidative aging of asphalt. The interior temperature was maintained at 100 °C for 20 h, and the pressure was set to 2.1 MPa. After the PAV aging, the asphalt samples were renamed MA-1-P, MA-2-P, etc.

#### 2.3.2. UV Aging Test

The annual average UV intensity of Urumqi in China was taken into consideration for simulating the UV aging of asphalt binders under practical pavement conditions. The asphalt binder was evenly poured onto a sample plate with a diameter of 10.5 cm, and samples with a thickness of 1.00 ± 0.05 mm were prepared. Then, these samples were aged by UV irradiation with a spectrum of 365 nm at 60 °C for 30 days, using a UV lamp intensity of 120 W/m^2^. After UV aging, the asphalt samples were renamed MA-1-U, MA-2-U, etc.

### 2.4. Microstructure Tests

#### 2.4.1. SEM (Scanning Electron Microscopy) Test

A small piece of GO/SBS-MA was selected and a thin conductive coating was used to cover the surface of the sample before testing. Hitachi (Tokyo, Japan) S-4800 scanning electronic microscopy (SEM) was employed for making observations, and its acceleration voltage in the test was 15 kV.

#### 2.4.2. FTIR (Fourier-Transform Infrared Spectroscopy) Test

In order to investigate the variation in the chemical composition of the asphalt binder before and after aging, a Brooke infrared spectrometer was utilized to scan the asphalt samples at room temperature. The FTIR spectra were taken in the range of 400–4000 cm^−1^ with a spectral resolution of 4 cm^−1^.

#### 2.4.3. AFM (Atomic Force Microscope) Test

An AFM (atomic force microscope) test was employed to study the changes in the micromorphology of the asphalt surface before and after aging [29]. For the preparation of AFM samples, a drop of asphalt binder was dripped onto a glass slide. After that, the glass slide was placed on a horizontal plate and left in the oven at 160 °C for 15 min, which allowed the asphalt to flow naturally and form a relatively level surface.

During the test, the elastic constant of the probe was adjusted to 0.4 N/m, and the scanning area was set to 20 cm × 20 cm. The image had a resolution of 512 × 512 pixels. For each observation, more than three parallel experiments were undertaken, and typical morphological results were chosen to avoid inadvertent errors in the data.

### 2.5. Rheological Property Tests

#### 2.5.1. BROOKFIELD Viscosity Test

In the BROOKFIELD viscosity test, an S-27 spindle was employed to test the viscosity of the asphalt binders. Usually, for different GO/SBS-compound-modified asphalts, the range of torques is from 10 to 98% and the values of shear rates are from 10 1/sec to 30 1/sec at 135 °C.

#### 2.5.2. DSR (Dynamic Shear Rheometer) Test

The DSR (dynamic shear rheometer) test according to ASTM D7175 [28] was used to characterize the rheological and viscoelastic behaviours of the asphalt binder. During the test, 25 mm parallel plates were chosen, with a 1 mm gap between them. The load frequency was set to 10 rad/s (1.59 Hz), the strain control was 10%, the temperature range was from 46 °C to 82 °C, and the temperature sweep interval was 6 °C.

#### 2.5.3. BBR (Bending Beam Rheometer) Test

In accordance with ASTM D6648 [28], a BBR (bending beam rheometer) test was used to determine the rheological properties of the GO/SBS-MA at low temperatures. Standard trabecular specimens of asphalt binders were prepared. Then, the specimens were placed in anhydrous ethanol liquid and maintained at −12 °C or −18 °C for 1 h. After that, a constant force (980 mN) was loaded onto the specimens, and the creep stiffness (S) and creep rate (m) of the asphalt binders were obtained at loading times of 60 s.

## 3. Results

### 3.1. Physical Properties of GO/SBS-MA

The test results of the physical properties are shown in Figure 2. It can be seen that the penetration and ductility of the GO/SBS-MA declined, while the softening point increased with an increase in the GO content. These results revealed that an increase in the GO content improved the high-temperature performance of the asphalt binder, while the low-temperature performance became slightly inferior. Several studies have pointed out that van der Waals interactions are expected to connect the asphalt molecules with the active groups on the surface of GO, and there were chemical effects between the asphalt binder and GO as well [30]. These physical and chemical functions enhanced the connection between GO/SBS and the asphalt binder, improving the stability of the GO/SBS-MA. Therefore, the effect of GO on the high-temperature performance of the asphalt binder was positive. On the contrary, the ductility decreased with an increase in the GO content, which can possibly be attributed to the fact that GO restrained the deformation of SBS and then affected the plasticity of the asphalt binder. Similarly, GO also increased the viscosity of the GO/SBS-MA. Except for the ductility, the physical properties of the GO/SBS-MA changed slowly when the content of GO increased, especially as the content increased from 1.0 to 1.5%, implying that there can be a relatively reasonable content of GO in GO/SBS composites.

### 3.2. Microstructure Analysis

#### 3.2.1. SEM Analysis

Graphene oxide (GO) is a derivative of graphene that appears as a brownish yellow powder with a high specific-surface free energy. It is a single or multilayer product formed by the stripping of graphite oxide, which has a typical quasi-two-dimensional structure. After blending with SBS, as can be seen in Figure 3, graphene oxide is dispersed in the block polymer and it forms an interface with SBS, thus restricting the movement of polymer chains and increasing the thermal degradation temperature, which effectively improves the thermal stability of the polymer. In addition, when graphene oxide is dispersed in SBS, graphene sheets can form a favourable barrier, preventing oxygen molecules from penetrating into the SBS and enhancing its resistance to thermal oxidative aging.

#### 3.2.2. FTIR Analysis

As shown in Figure 4a, the characteristic peaks of the original GO/SBS-MA were mostly found at 2920 cm^−1^, 2850 cm^−1^, 1600 cm^−1^, 1455 cm^−1^, 1375 cm^−1^, 1032 cm^−1^, 966 cm^−1^, and 699 cm^−1^ before aging, whereas after aging, the newly added characteristic peaks were found at 1699 cm^−1^ and 1173 cm^−1^. Information on the functional groups that correspond to these characteristic peaks is shown in Table 5.

In order to quantitatively evaluate the differences in the chemical composition of GO/SBS-MA before and after aging, the areas of the characteristic peaks of the carbonyl group near 1699 cm^−1^, the sulfoxide group near 1030 cm^−1^, and polybutadiene near 966 cm^−1^ were calculated. To eliminate the influence of other factors on the calculation results, the areas of the peak near 2800–3000 cm^−1^ were also calculated for determining the aging indexes of the FTIR analysis. Equations (1)–(3) were used to calculate the carbonyl index, sulfoxide index, and butadiene index, respectively:(1)$$I_{C=O}=\frac{A_{1699}}{\sum A}$$
(2)$$I_{S=O}=\frac{A_{1032}}{\sum A}$$
(3)$$I_{C=C}=\frac{A_{966}}{\sum A}$$
where A is the area of the specific absorption peak and ∑A is the sum of the peak areas at 2800–3000 cm^−1^. The calculation outcomes of different indexes are shown in Figure 5.

It is acknowledged that the aging process of asphalt binders mainly comprises a few free radical reactions with the participation of oxygen, which primarily involves four periods: the initiation, growth, disproportionation, and termination of the molecular chain [32]. The reaction processes are illustrated by the following chemical equations:

Initiation steps:(4)RH→R·+H·
(5)$$RH+O_{2}\to R\cdot +HOO\cdot$$
(6)$$R\cdot +O_{2}\to ROO\cdot$$

Growth and disproportionation steps:(7)ROO·+RH→ROOH+R·
(8)ROOH→RO·+HO·
(9)$$ROOH+RH\to RO\cdot +RH\cdot +H_{2}O$$
(10)RO·+RH→ROH+R·

Termination steps:(11)R·+R·→R−R
(12)R·+ROO·→ROOR
(13)$$ROO\cdot +ROO\cdot \to ROOR+O_{2}$$

Obviously, the alkoxy radical bond is broken and the unstable hydroperoxide intermediates are degraded in the aging process of asphalt binders, leading to the generation of carbonyl functional groups [33].

As far as carbonyl indexes are concerned, the values of the indexes after UV aging were higher than those after PAV aging, which was attributed to the hydrogen capture process of the peroxide radical in the polymers, which could be considerably accelerated under the influence of UV irradiation [34].

As shown in Figure 5, even though aging caused an increase in the carbonyl indexes, the GO/SBS-MA with a higher GO content had lower carbonyl indexes, obviously. And not only that, but the higher GO content played an obvious role in preventing the fracture and oxidation of the double bonds in SBS, especially in UV aging. Similar to carbon black, GO also absorbed UV radiation, resulting in the free reaction of SBS and the asphalt binder being successfully impeded.

It has been acknowledged that the sulfoxide indexes of GO/SBS-MA increase with a higher degree of aging. For MA-2, the values of the sulfoxide indexes were reduced by 42.2% after PAV aging and 35.4% after UV aging. And there was no doubt that GO also restricted the formation of sulfoxide groups. Recent studies have shown that GO can form an intercalation structure in asphalt binders [22]. Accordingly, GO can generate this structure in SBS segments and prevent oxygen from diffusing through SBS and asphalt, leading to a reduction in the sulfoxide indexes [35]. As for the butadiene index, the reduction rate in the butadiene index declined with an increase in the GO content, especially in the UV test, illustrating that there was an inhibitory effect of GO on SBS degradation.

In addition to the above-mentioned reasons, the abundant functional group in GO can generate free radicals, which would act as hydrogen donors to capture reactive radicals, such as hydroperoxides produced by SBS and asphalt during aging. After that, the reactive radicals would be transformed into relatively stable compounds and the chain reaction would be terminated consequently, contributing to the anti-aging properties of GO/SBS-MA.

In conclusion, the FTIR analysis of the GO/SBS-MA demonstrated that increasing the content of GO could prevent the production of carbonyl and sulfoxide groups and restrain the breakdown of C=C bonds in the polybutadiene (PB) segment, owing to the inhibiting effect of GO on the aging of asphalt binders. Figure 6 briefly illustrates the anti-aging mechanism of the GO/SBS-MA.

#### 3.2.3. AFM Analysis

In the AFM test, the max height and roughness were employed to evaluate the surface morphology of the GO/SBS-MA before and after aging, as can be seen in Table 6. Regardless of whether PAV or UV aging was used, both indexes decreased. However, as the content of GO increased, the quantity of variation significantly reduced for both indexes.

As shown in Figure 7, an increase in the GO content in the GO/SBS-MA had a significant influence on the micromorphology of the asphalt binder before and after aging. As for the unaged asphalt binder, there were no significant differences in the “bee” structures for four asphalt binders. It has been reported that the formation of “bee” structures is mainly related to the condensation process of resins and asphaltenes [36]. In all the asphalt binders, the “bee” structures varied in size and their distributions were random in local areas. Interestingly, more “bee” structures came out with an increase in the GO content. One possible explanation is that asphaltenes with a strong polarity have a tendency to aggregate together and form the assorted morphology of the “bee” structures affected by GO. Furthermore, a physical blend between GO/SBS and an asphalt binder accelerated the aggregation of asphaltenes. The crosslinked network structure of SBS in the asphalt binder prevented the migration and aggregation of asphaltenes, but GO particles also served as aggregation agents and promoted the heterogeneous aggregation of asphaltenes.

Admittedly, the microstructure of an asphalt binder is highly related to its components and their thermal history [37]. For aged asphalt binders, the “bee” structures became blurred or nearly undistinguishable on the surface of the GO/SBS-MA. This variation was detected more easily after the PAV aging compared with the UV aging. Moreover, there were more significant distinctions in the microstructure than could be seen in the asphalt binder with a higher content of GO. In addition, it is known that aging causes the migration of asphalt components in such a way that the light compositions evolve into the heavy compositions. Specifically, the non-polar molecules are gradually translated into resins and asphaltenes. A study by Yu et al. [32] discovered that aging can damage the network structure of the polymer-rich phase and exacerbate the competition between the polymer and asphaltenes. In addition, the new equilibrium in the asphalt binder was not conducive to the development of the polymer-rich phase. Precisely, the aggregation of asphaltenes was amplified by aging-related variations in the polymer-rich phase and the asphalt components, and the higher the content of GO, the more obvious was the homogeneous aggregation of asphaltenes.

As shown in Figure 7, the needle-like aggregates on the surface of the GO/SBS-MA after aging were more than those resulting from unaged asphalt. Apart from that, the distributions of the needle-like aggregates were more homogeneous with an increase in the GO content, just like the images of MA-3-P, MA-4-P, MA-3-U, and MA-4-U. These findings indicated that GO had a significant influence on the aggregated structure on the surface of GO/SBS-MA and promoted the stability of the asphalt binder, which was in line with the conclusions of the FTIR analysis.

### 3.3. Rheological Properties

#### 3.3.1. DSR Analysis

In the DSR test, a temperature sweep test was applied to detect the rheological properties of the SBS-MA that varied with temperature. As shown in Figure 8a, the complex modulus (G*) of the unaged asphalt increased with an increase in the GO content. And there were a few smaller differences among all the asphalt binders as the temperature rose. Owing to the good dispersity of GO in GO/SBS-MA, GO had a certain thickening effect on the asphalt binder, and the higher the GO content was, the more noticeable the effect was. There was no doubt that the complex modulus (G*) of all the asphalt samples increased with the aggravation of aging. Regardless of PAV or UV aging, MA-1 exhibited the highest complex modulus (G*), whereas MA-4 demonstrated the lowest. For all the asphalt binders, the increasing trend of the complex modulus (G*) was opposite before and after aging. Undeniably, it was suggested that GO possesses a positive effect on asphalt aging resistance. Therefore, the delaying of asphalt aging could be achieved through a reasonable increase in the content of graphene oxide (GO) for GO/SBS-MA composites.

The phase angle (δ) represents the ratio of the viscous component to the elastic component in an asphalt binder [37]. As for unaged GO/SBS-MA, the higher the GO content, the smaller the phase angle. Additionally, it was manifested that the addition of GO made the asphalt binder more elastic, which was beneficial for the high-temperature performance of the asphalt binder. Unquestionably, the phase angle (δ) decreased with the deepening of the aging degree. As shown in Figure 8b, the phase angle (δ) showed an increasing trend with a rise in the temperature and a decrease after aging, especially PAV aging. It was evident that the phase angle (δ) of MA-1-P was reduced by 14.0% at 46 °C, while the phase angle (δ) of MA-4-P was only reduced by 4.87%. In conclusion, the higher content of GO in the asphalt binder had an advantage in restraining the transformation of the viscoelastic composition of the GO/SBS-MA.

Rutting factors (G*/sinδ) were used to evaluate the high-temperature rheological properties of all the asphalt binders. And there is no doubt that the higher the value of the rutting factor, the better the high-temperature performance of the GO/SBS-MA. As shown in Figure 9, an increase in the GO content definitely contributed to the high-temperature performance of unaged asphalt binders. Regardless of the PAV aging or UV aging test results, the asphalt binders had higher rutting factors compared to the unaged GO/SBS-MA. To sum up, there is no denying the fact that asphalt binders with a higher content of GO not only possess a better high-temperature performance, but they also reduce the adverse impact caused by aging.

#### 3.3.2. BBR Analysis

It was anticipated that the asphalt binders with a better low-temperature performance would have a lower creep stiffness and a higher creep rate. As shown in Figure 10, the unaged GO/SBS-MA with a higher content of GO possessed a higher creep stiffness (S) and a lower creep rate (m), implying that GO had a slightly negative impact on the low-temperature performance of asphalt binders. This is consistent with the aforementioned results for low-temperature ductility. And it is acknowledged that an improvement in the low-temperature performance of SBS-MA mainly relies on the flexibility offered by the PB (polybutadiene) segments. Due to the barrier effect of GO on the partial movement of the PB segments, there was marginal impairment of the low-temperature properties.

For the aged asphalt binders, as an example, the creep stiffness (S) of MA-2-P increased by 21.3% and the creep rate (m) decreased by 11.9% at −12 °C, while the creep stiffness (S) of MA-4-P increased by 3.4% and the creep rate (m) decreased by 5.9% compared with an unaged asphalt binder. Apparently, GO/SBS-MA with a higher content of GO had a lower stiffness modulus (S) value and a higher creep rate (m) value after aging. This indicates that GO had an advantageous influence on preventing the aging of GO/SBS-MA. When the test temperature shifted from −12 °C to −18 °C, the creep stiffness (S) increased and the creep rate (m) decreased. The GO/SBS-MA was close to the glassy state in this condition, and the molecular segments of the asphalt binders nearly failed to move. It was clear that aging caused the asphalt binder to gradually harden at lower temperatures, weakening the low-temperature properties of the GO/SBS-MA. Certainly, asphalt binders with a higher content of GO have a smaller effect on decreasing the low-temperature performance. Therefore, although GO caused a disadvantageous effect on the low-temperature performance of asphalt binders, it had a beneficial influence on inhibiting asphalt aging.

In addition, the changing rate of key indexes before and after aging can be seen in Figure 11. After PAV and UV, when the content of GO increased from 0% to 1.5%, the increase in the changing rate for the sulfoxide index (Is=o) was reduced by 132% and 86% and the decrease in the changing rate for the butadiene index (Ic=c) was reduced by 8% and 22%, respectively. Moreover, the increase in the changing rate for the complex modulus (64 °C) was reduced by 253% and 219% and the increase in the changing rate for the creep stiffness (−12 °C) was reduced by 16% and 20%, respectively. In conclusion, the incorporation of GO led to a reduction in the margin of aging-related key indexes. Hence, GO exhibited a favourable efficacy in mitigating SBS degradation and delaying asphalt aging.

## 4. Conclusions

In this paper, GO/SBS composites were prepared via the solution-blending method. These GO/SBS composites with varying GO contents were employed to prepare GO/SBS-MA. The changes in the macroscopic performance and the microstructure of GO/SBS-MA before and after aging were analysed, respectively. And the effects of GO/SBS on the anti-aging performance of asphalt binders were investigated. The following conclusions can be made:According to the physical properties, the penetration and ductility of the GO/SBS-MA declined and the softening point and viscosity increased with increasing GO content. The results revealed that the increase in the GO content improved the high-temperature performance of the asphalt binder, while the low-temperature performance became slightly inferior.According to the FTIR analysis, increasing the content of GO could prevent the production of carbonyl and sulfoxide groups and restrain the breakdown of C=C bonds in the polybutadiene (PB) segment. The results indicated that GO could inhibit the degradation of SBS and the aging of asphalt binders.From the AFM analysis, it was observed that the needle-like aggregates on the surface of the GO/SBS-MA after aging were more than that of unaged asphalt. The distributions of the needle-like aggregates were more homogeneous with an increase in the GO content. These findings show that GO had a significant influence on the formation of the aggregated structure and that it enhanced the high-temperature stability of the asphalt binder.According to the rheological properties, asphalt binders with a higher content of GO not only exhibit a better high-temperature performance, but they also undermine the adverse impact caused by aging, while GO causes a disadvantageous effect on the low-temperature performance of asphalt binders.

## Figures and Tables

**Figure 1 polymers-16-01504-f001:**
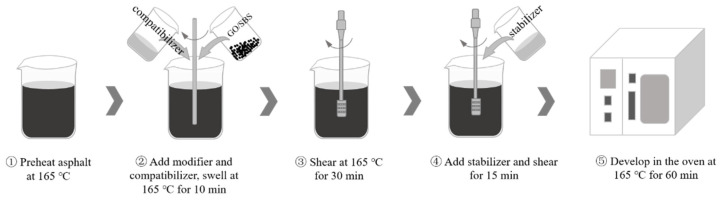
Preparation of GO/SBS-MA.

**Figure 2 polymers-16-01504-f002:**
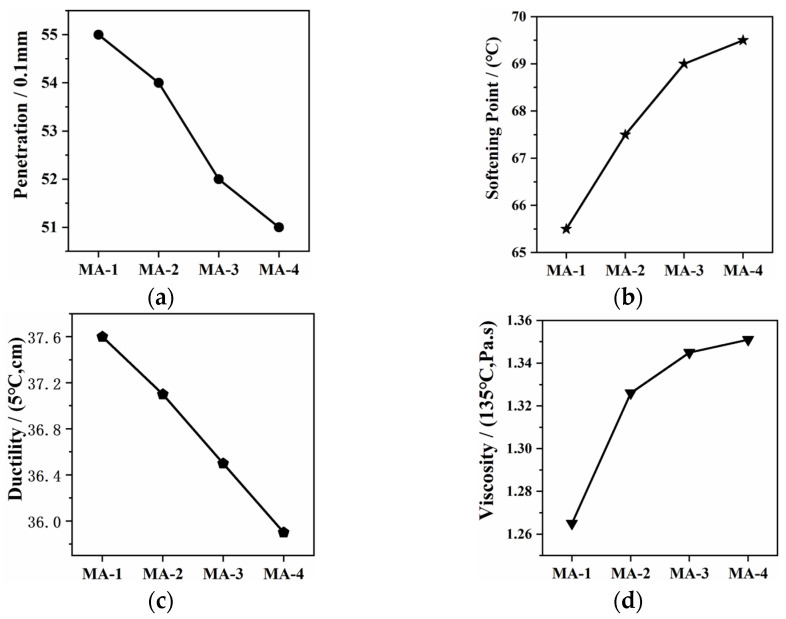
Physical properties of GO/SBS-MA. (**a**) Penetration; (**b**) softening point; (**c**) ductility; and (**d**) viscosity.

**Figure 3 polymers-16-01504-f003:**
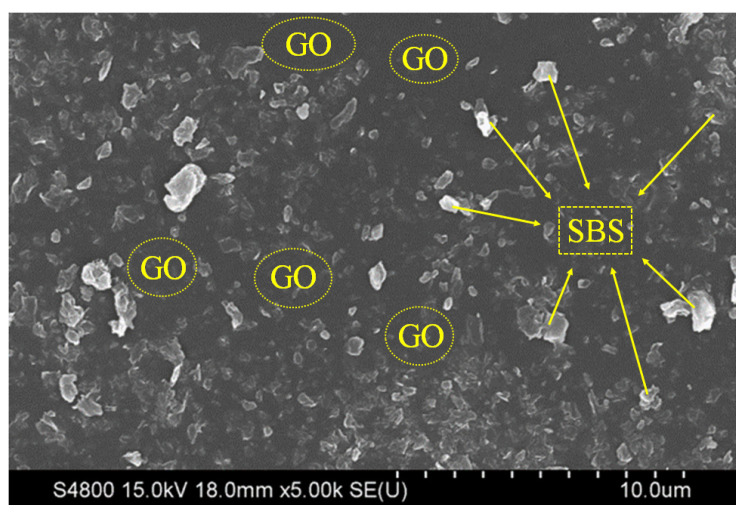
SEM photograph of GO/SBS-MA.

**Figure 4 polymers-16-01504-f004:**
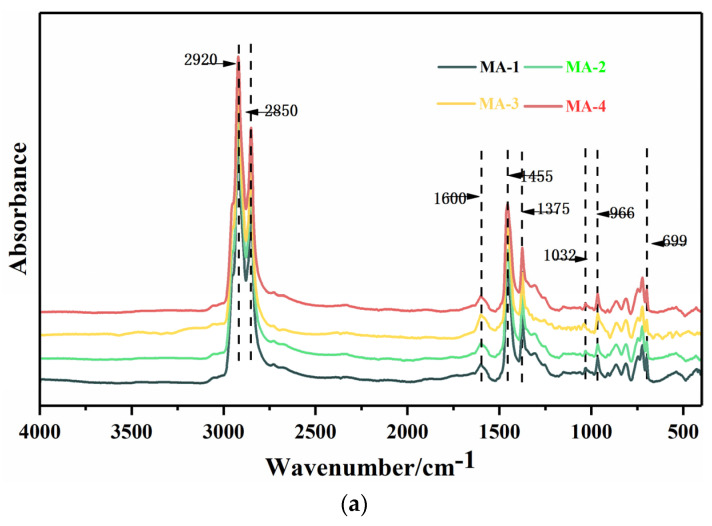
Infrared spectrogram of GO/SBS-MA. (**a**) Infrared spectrogram of GO/SBS-MA before PAV aging; (**b**) infrared spectrogram of GO/SBS-MA after PAV aging; and (**c**) infrared spectrogram of GO/SBS-MA after UV aging.

**Figure 5 polymers-16-01504-f005:**
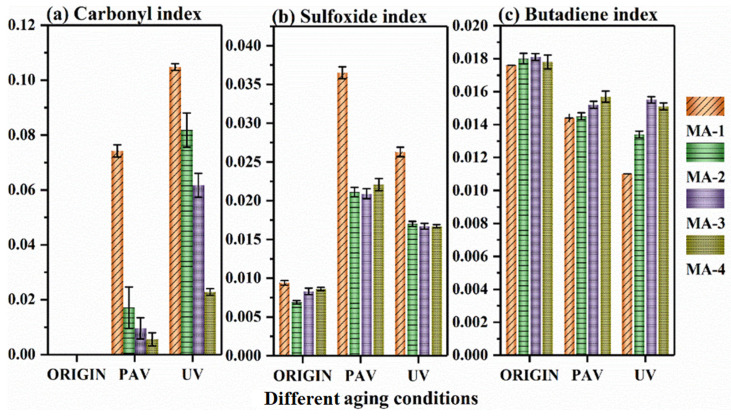
Indexes before and after different aging methods.

**Figure 6 polymers-16-01504-f006:**
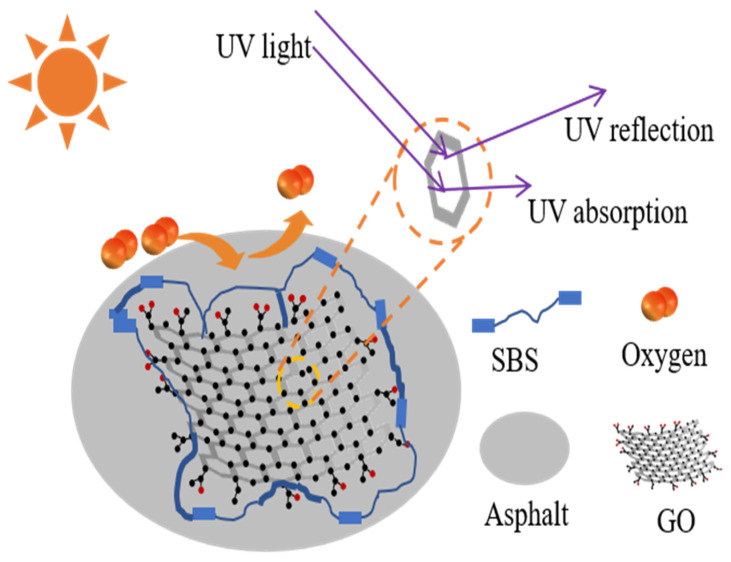
Anti-aging mechanism diagram of GO/SBS-MA.

**Figure 7 polymers-16-01504-f007:**
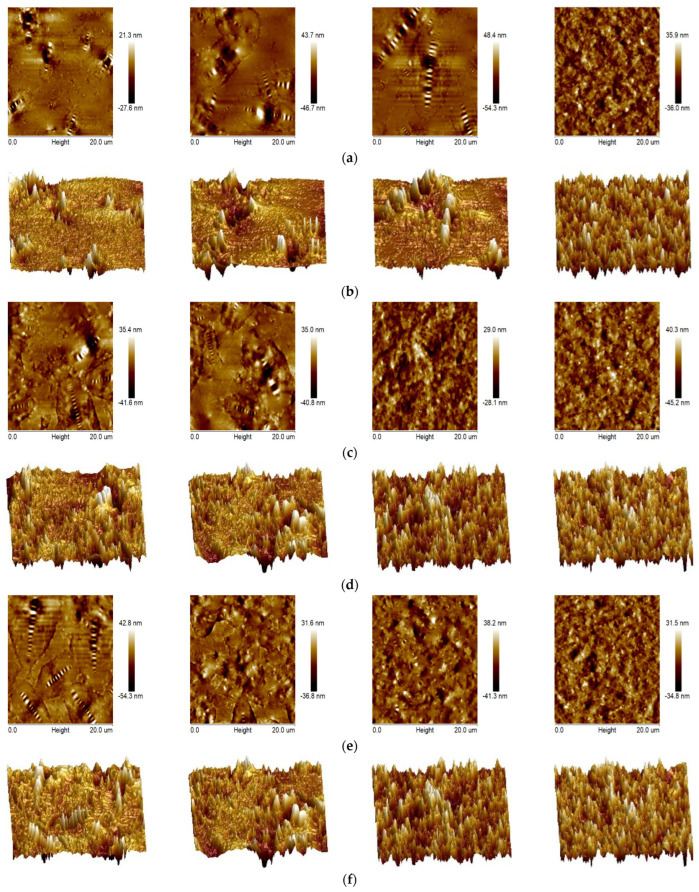
AFM images of GO/SBS-MA; (**a**) 2D AFM images of MA-1, MA-2, MA-3, and MA-4; (**b**) 3D AFM images of MA-1, MA-2, MA-3, and MA-4; (**c**) 2D AFM images of MA-1-P, MA-2-P, MA-3-P, and MA-4-P; (**d**) 3D AFM images of MA-1-P, MA-2-P, MA-3-P, and MA-4-P; (**e**) 2D AFM images of MA-1-U, MA-2-U, MA-3-U, and MA-4-U; and (**f**) 3D AFM images of MA-1-U, MA-2-U, MA-3-U, and MA-4-U.

**Figure 8 polymers-16-01504-f008:**
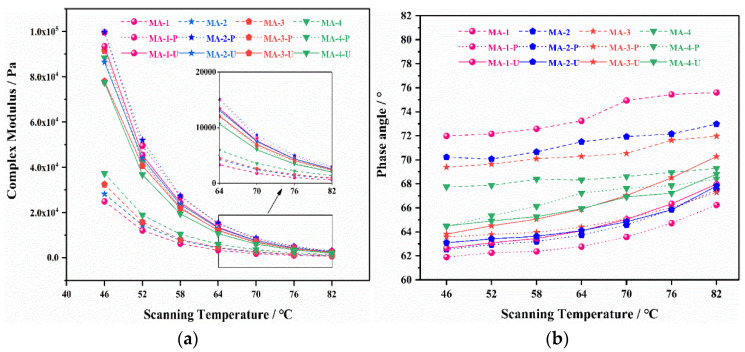
Viscoelastic properties of GO/SBS-MA. (**a**) Complex modulus (G*); (**b**) phase angle (δ).

**Figure 9 polymers-16-01504-f009:**
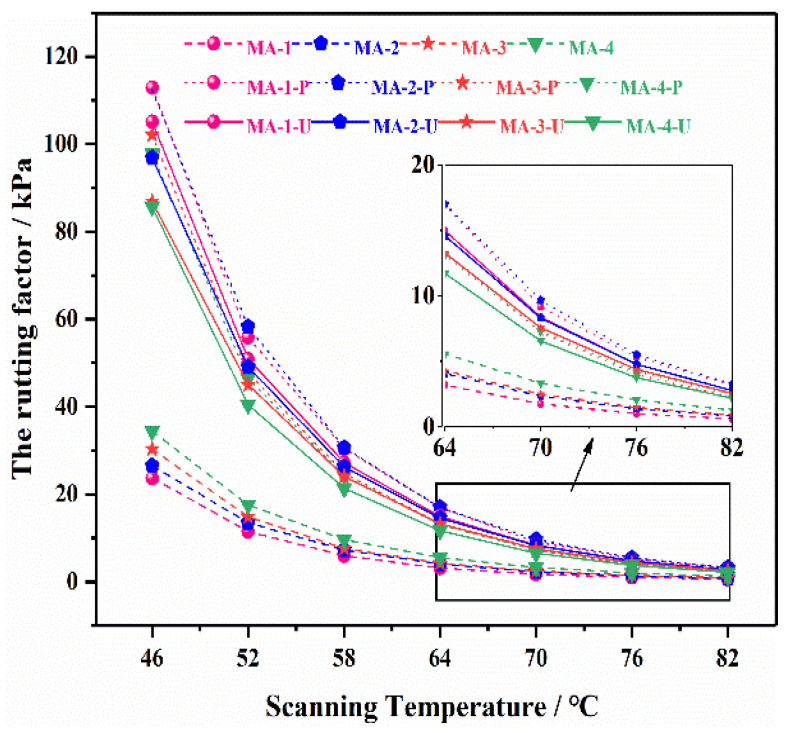
The rutting factor of GO/SBS-MA.

**Figure 10 polymers-16-01504-f010:**
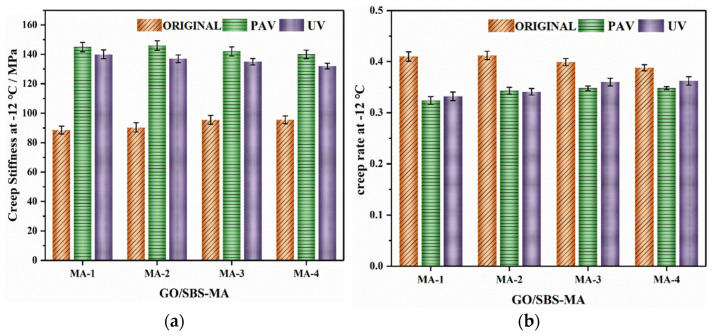
Creep stiffness (S) and creep rate (m) of GO/SBS-MA. (**a**) Creep stiffness at −12 °C; (**b**) creep rate at −12 °C; (**c**) creep stiffness at −18 °C; and (**d**) creep rate at −18 °C.

**Figure 11 polymers-16-01504-f011:**
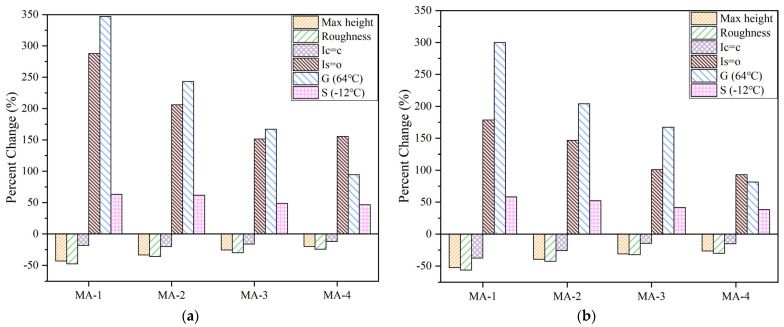
The changing rate of key indexes before and after aging. (**a**) GO/SBS-MA(PAV), (**b**) GO/SBS-MA(UV).

**Table 1 polymers-16-01504-t001:** Physical properties of A-70.

Property	Unit	Value	Standard [28]
Penetration (25 °C)	0.1 mm	72	ASTM D5-13
Softening point	°C	49.5	ASTM D36-76
Ductility (15 °C)	cm	>100	ASTM D113-99
Viscosity (135 °C)	Pa·s	0.535	ASTM D2170
Flash point	°C	284	ASTM D92
Solubility	%	99.6	ASTM D2042

**Table 2 polymers-16-01504-t002:** Basic parameters of GO.

Appearance	Purity	Number of Layers	Carbon Content	Oxygen Content	Lamellar Diameter
Black powder	98%	1~2	<85 wt.%	>7 wt.%	0.5~10 μm

**Table 3 polymers-16-01504-t003:** Technical properties of SBS modifier.

Technical Index	Test Result	Unit
Styrene content	31	Wt/%
Volatile content	0.38	Wt/%
300% tensile stress	2.41	MPa
Tensile strength	26.8	MPa
Elongation at break	735	%
Hardness	81	Shore A
Molecular weight (Mn)	135,783	g/mol

**Table 4 polymers-16-01504-t004:** Proportion of each material in modified asphalt.

Material	Weight (g)
Base asphalt	600.0
GO/SBS composites	30.0
Stabilizer	1.2
Compatibilizer	6.0

**Table 5 polymers-16-01504-t005:** Infrared absorption peaks and corresponding functional groups of GO/SBS-MA.

Position of Absorption Peak/cm^−1^	Functional Group [31]
2920, 2850	The asymmetric and symmetric stretches of CH2
1699	Carbonyl
1600	Conjugated double bond (benzene ring skeleton vibration)
1455, 1375	CH3 and CH2 bending vibration
1173	C-O in ether, ester, and anhydride
1032	Sulfoxide group
966	Polybutadiene segment
699	C-H on benzene ring of polystyrene segment
650~910	C-H on substituent of benzene ring

**Table 6 polymers-16-01504-t006:** The max height and roughness of GO/SBS-MA before and after aging.

GO/SBS-MA	After PAV Aging		After UV Aging	
	Max Height/%	Roughness/%	Max Height/%	Roughness/%
MA-1	−42.9	−47.6	−52.4	−56.2
MA-2	−33.3	−35.7	−39.3	−42.4
MA-3	−25.5	−29.8	−30.9	−32.1
MA-4	−20.0	−24.3	−26.3	−30.0

## Data Availability

The data are contained within the article.

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
