# Peer review of "The Effects of Aging on Microstructures and Rheological Properties of Modified Asphalt with GO/SBS Composite"

_polymers, 2024, doi:10.3390/polym16111504_

Round 1

Reviewer 1 Report

Comments and Suggestions for Authors

This manuscript looked at the impact of adding Graphene oxide (GO) into SBS modified asphalt specifically focused on the aging sensitivity. The polymer modified binder with various contents of GO has gone through long-term aging conditions. Laboratory tests including FTIR, AFM, DSR and BBR were performed to evaluate the effect from GO on the rheological properties of polymer modified binder. This study was well designed and structured. The findings from this work can shed light on addressing the high aging sensitivity from polymer modified binder. Detailed comments are as follows:

(1)  Abstract: It is recommended to add some sentences at the beginning of the abstract to provide a brief introduction and background leading to this study.

(2)  Line 121: Could the authors provide some justifications for the selection of GO portions?

(3)  Figure 1: what is the purpose of step 5 in the figure?

(4)  Table 4. It is normally assumed 100% for the total modified asphalt. The ratio is recommended to update the ratio numbers for the components in the table.

(5)  The discussion on Figure 4 focused on Carbonyl index and Sulfoxide index. It is suggested to add more analysis on Butadiene index.

Reviewer 2 Report

Comments and Suggestions for Authors

The article by Xie H. et al. describes bitumen binders containing graphene oxide and styrene-isoprene-styrene block copolymer (SBS) as modifiers. The authors obtain graphene-containing SBS through its solution, add the resulting nanocomposite into bitumen to obtain three different composite bitumens, and then investigate their rheological properties, morphology, and IR spectra before and after thermal and UV aging. As a result, the authors show a positive effect of graphene on rutting and aging resistances. The article is well written and can be published after some improvements.

Specific comments are as follows.

Title. Corrections needed, e.g., “The effects of aging on a microstructure and rheological properties of an asphalt modified with GO/SBS composite”.

Line 48: “Isophorone diisocyanate-layered double hydroxide”. It is unclear what this compound is. Hydroxide of what? What element? Aluminum, silicon, titanium?

Table 3. The molecular weight of SBS must be specified.

Line 114: “Firstly, GO powder and SBS were blended into the cyclohexane solution by using a mixer”. It is necessary to specify the concentration of SBS in cyclohexane, at least approximately.

Line 126: “named MA-2, MA-3, and MA-4”. The mass fractions of GO in parentheses need to be given.

Table 4. The Ratios of Materials (100%, 5%, 0.2%, 1%) need to be recalculated to make their sum equal to 100%. The chemical structures of the stabilizer and compatibilizer and the purpose of their introduction into the asphalt must be written.

Line 197: “GO could also increase the viscosity”. The conditions for viscosity measurement (specific shear rate or shear stress) should be written in the Materials and Methods section.

Figure 2. It is unclear why the authors do not provide data for samples after their thermal and ultraviolet aging. It is very interesting how viscosity, penetration, ductility, and softening point change because of aging and how graphene oxide prevents their change. The authors claim that graphene oxide prevents aging, and presenting this data is a great way to demonstrate this.

Table 5. Supporting references are needed where the matching of absorption bands to the specific functional groups is shown (e.g., FTIR spectra of various bitumens and asphalts have previously been studied in detail in 10.3390/molecules28052065).

Lines 301, 316, 318, 320, 322, 430-433: “GO particles also served as a nucleating agent and promoted heterogeneous nucleation of asphaltenes”, “homogeneous nucleation of asphaltenes”, “the needle-like crystals”. I am not sure that the term "nucleation" is a good choice, as it implies the crystallization of asphaltenes. However, asphaltenes do not crystallize (judging from diffractograms of asphalts). Perhaps it would be better to replace “nucleating agent” with “aggregation agent”, “heterogeneous nucleation of asphaltenes” with “heterogeneous aggregation of asphaltenes”, “the needle-like crystals” with “the needle-like aggregates”, and so on.

Figures 7a and 8: Probably, these figures are better represented in semi-logarithmic coordinates. In this case, the difference between the samples will be more evident at high temperatures, and it will not be necessary to use magnifying insets.

Figure 10. It is unclear what is meant by "Ic=c" and "Is=0" in the legends. In addition, this figure is worth discussing in more detail.

Comments on the Quality of English Language

The English language requires extensive editing, especially verb tenses.

Round 2

Reviewer 2 Report

Comments and Suggestions for Authors

The authors have improved the article for its publication.

Comments on the Quality of English Language

The English language requires moderate editing.